# Effects on Corticospinal Tract Homology of Faremus Personalized Neuromodulation Relieving Fatigue in Multiple Sclerosis: A Proof-of-Concept Study

**DOI:** 10.3390/brainsci13040574

**Published:** 2023-03-29

**Authors:** Massimo Bertoli, Angela Tataranni, Susanna Porziani, Patrizio Pasqualetti, Eugenia Gianni, Joy Grifoni, Teresa L’Abbate, Karolina Armonaite, Livio Conti, Andrea Cancelli, Carlo Cottone, Franco Marinozzi, Fabiano Bini, Federico Cecconi, Franca Tecchio

**Affiliations:** 1Laboratory of Electrophysiology for Translational neuroScience (LET’S), Laboratory of Agent-Based Social Simulation (LABSS), Italian National Research Council (CNR), Via Palestro 32, 00185 Rome, Italy; 2Department of Neuroscience, Imaging and Clinical Sciences, University ‘G. D’Annunzio’ of Chieti-Pescara, 66100 Chieti, Italy; 3Department of Mechanical and Aerospace Engineering, “Sapienza” University of Rome, 00185 Rome, Italy; 4Department of Public Health and Infectious Diseases, Sapienza University of Rome, 00185 Rome, Italy; 5Unit of Neurology, Neurophysiology, Neurobiology, Department of Medicine, Università Campus Bio-Medico di Roma, 00128 Rome, Italy; 6Engineering Faculty, International Telematic University Uninettuno, 00186 Rome, Italy; 7Istituto Nazionale di Fisica Nucleare, Sezione Roma Tor Vergata, 00133 Rome, Italy

**Keywords:** transcranial direct-current stimulation (tDCS), transcranial electric stimulation (tES), transcranial magnetic stimulation (TMS), corticospinal tract, multiple sclerosis (MS), precision medicine

## Abstract

Objectives: Fatigue in multiple sclerosis (MS) is a frequent and invalidating symptom, which can be relieved by non-invasive neuromodulation, which presents only negligible side effects. A 5-day transcranial direct-current stimulation, 15 min per day, anodically targeting the somatosensory representation of the whole body against a larger occipital cathode was efficacious against MS fatigue (fatigue relief in multiple sclerosis, Faremus treatment). The present proof-of-concept study tested the working hypothesis that Faremus S1 neuromodulation modifies the homology of the dominant and non-dominant corticospinal (CST) circuit recruitment. Methods: CST homology was assessed via the Fréchet distance between the morphologies of motor potentials (MEPs) evoked by transcranial magnetic stimulation in the homologous left- and right-hand muscles of 10 fatigued MS patients before and after Faremus. Results: In the absence of any change in MEP features either as differences between the two body sides or as an effect of the treatment, Faremus changed in physiological direction the CST’s homology. Faremus effects on homology were more evident than recruitment changes within the dominant and non-dominant sides. Conclusions: The Faremus-related CST changes extend the relevance of the balance between hemispheric homologs to the homology between body sides. With this work, we contribute to the development of new network-sensitive measures that can provide new insights into the mechanisms of neuronal functional patterning underlying relevant symptoms.

## 1. Introduction

### 1.1. MS Fatigue 

Multiple sclerosis (MS) is a chronic neurodegenerative disease, mainly driven by inflammatory processes, resulting in demyelinating lesions detectable in the white and gray matter throughout the central nervous system [1]. The onset of the disease typically ranges from 20 to 40 years of age and, in its most common clinical manifestation known as relapsing–remitting MS (RRMS), occurs with acute reversible neurological deficits lasting several days or weeks, while in a minority of patients, there is a gradual progression of the disease known as primary progressive MS (PPMS). Whatever the clinical manifestation and stage of the disease, chronic fatigue represents one of the most troubling symptoms impairing the daily life of up to 80% of the patients [2]. Defined as "the decrease in physical and/or mental performance that results from changes in central, psychological, and/or peripheral factors" [3,4], fatigue poses a challenge both for its diagnostic assessment and for its treatment, which is still limited to drugs, with little evidence of effectiveness while also showing unpleasant side effects [5].

### 1.2. Why Applying Excitatory Neuromodulation over S1 in Treating MS Fatigue 

Considering the functional neural features of MS fatigue [6], there is consensus in the literature concerning a widespread alteration of neuronal electrical activity at the level of the sensorimotor network [7]. In particular, MS fatigue patients display a depleted excitability of primary somatosensory and post-central networks [8,9,10,11] in contrast to a hyperactivation and excessive excitability observed in frontal areas [12] and primary motor cortices (M1) both at rest and during motor execution [13,14,15]. Therefore, chronic fatigue, together with a plethora of diseases and symptoms stemming from an altered neuronal electrical activity, stands out as a suitable target of therapeutic interventions by means of electroceuticals [16,17], a new emerging class of devices that treat ailments through the supply of electrical currents also via invasive or non-invasive neuromodulations. Among the non-invasive ones, transcranial direct-current stimulation (tDCS) is capable of modulating the membrane excitability of neurons belonging to wide cortical networks by delivering weak electric currents through electrodes positioned on the scalp, ultimately inducing a change in cortical excitability when delivered for sufficient time [18,19,20,21]. tDCS has been extensively used to treat MS fatigue [22,23], proving its appropriateness in dealing with functional alterations underlying this symptom with respect to other neurophysiological techniques with more focal mechanisms of action, such as repetitive transcranial magnetic stimulation (rTMS), that have been proved to be of limited application in this condition [24]. 

In line with the aforementioned literature on brain electrical activity features of MS fatigue resulting in an impaired parieto-frontal functional connectivity, we adapted a tDCS schema that had already achieved an enhancement of resistance to fatigue in healthy people [25] implementing an electroceutical intervention against fatigue in MS called Faremus (fatigue relief in multiple sclerosis) [23,26,27,28]. 

Faremus is a personalized 5-day anodal tDCS (1.5 mA, 15 min per day) that targets the whole-body primary somatosensory area (S1) via a documented procedure of shaping the anodal electrode (35 cm^2^) based on MRI-derived individual cortical folding of the central sulcus in order to leave the primary motor cortex (M1) and pre-central areas unstimulated [29,30,31]. The cathode over bilateral occipital sites is a double area electrode (7 × 10 cm^2^) in order to avoid inhibitory effects on visual cortices.

Fatigue in MS has been targeted by a variety of intervention protocols employing tDCS, with different stimulation parameters and cortical targets [32,33]. Among these, the dorsolateral pre-frontal cortex, M1, and S1 represent the most investigated. However, differences due to heterogeneity in the number of sessions, current intensity and duration, sample size, and outcome measure do not allow disambiguating which tDCS parameters are preferable. Nevertheless, the results of a recent quantitative and meta-analysis review by Gianni et al. [23] focusing on class 1 RCTs involving tDCS interventions against symptoms mainly related to neuronal electrical unbalances revealed that Faremus bilateral S1 anodal stimulation consistently showed clinical efficacy in specifically relieving fatigue [23,26,27,28].

### 1.3. Balance between Hemilateral Homologs Is Critical for Functional Ability

Healthy brain functioning relies on the balance between hemispheric homologous areas, enabled by the core cerebral mechanism of inter-hemispheric inhibition, an ubiquitous feature of brain functioning implemented by crossed facilitation projections affecting surround/lateral inhibition networks aimed at supporting contrast-enhancing and integrative functions through the balancing of the activity of the two hemispheres [34]. The interplay between the activity of homologous hemispheric areas can thus be seen as an ubiquitous structural–functional mechanism that supports the plastic adaptation and learning processes of the brain [35,36] entering the regulation of inhibition and excitation mechanisms that support the functionality of the body segment they control.

As previously observed, MS fatigue patients display an electrophysiological profile compatible with local alterations of physiological excitatory–inhibitory mechanisms [7,13,14,15], secondary or giving rise to fatigue itself, that also reflect into the imbalances of the inter-hemispheric functional relationship between sensorimotor regions both at rest [37] and during movement execution [38,39,40]. These pieces of evidence pairing increased fatigue symptoms with an altered dynamic interplay between homologous cortical areas, in the absence of structural parenchymal changes, further support the primarily functional alteration of the sensorimotor system in MS fatigue. Notably, the somatosensory areas involved in the genesis and treatment of fatigue by means of Faremus are part of the origin of the CSTs. In fact, about 60% of the fibers that make up the CST come from the pre-central gyrus [41] and about 30% from the primary somatosensory areas [42] and parietal operculum [43]. Moreover, a previous study on the mechanisms of action of Faremus treatment [11] had shown that the reduction in fatigue levels was associated with a rebalancing of the functional connectivity within the key nodes of the sensorimotor network (bilateral S1 and M1), resulting in a main reinforcement of the interplay between the two homologous M1 areas.

On these bases, it is conceivable to expect that the effects of Faremus S1 neuromodulation, aimed at normalizing the activation patterns within the primary sensorimotor system, can be detectable along the whole central–peripheral CST pathways as expressed by TSM-induced MEP features [44,45].

### 1.4. Corticospinal Tracts and MEP Morphology

Single-pulse supra-threshold TMS over M1 is widely used to assess the functionality of the CST as a result of pulse propagation along the spinal cord detected at the contralateral muscle level with surface electrodes as the motor-evoked potential (MEP) [46].

Within our aim of assessing inter-side (dominant vs. non-dominant) homology, we derived the measure as the similarity of circuit recruitment patterns as we did for somatosensory evoked potentials (SEPs) [47] and fields (SEFs) [48,49] via the morphologies of the evoked response, MEPs in the present investigation. Analogously, and in agreement with recent large-scale assessments of MEP morphology [50], here, we aimed at quantifying how Faremus treatment modifies the balance of the CST recruitment between dominant and non-dominant body sides as assessed by the morphology similarity of left- and right-hand muscle MEPs. To this end, we used the measure of similarity between two curves, the Fréchet distance [51,52], as we already did in healthy volunteers [53]. 

### 1.5. Study Aim

By the present proof-of-concept study, we pose the working hypothesis that Faremus will modify in the physiological direction the homology between the two CSTs. We will also evaluate the relationship of the inter-side MEP morphology similarity with the intra-side for both the dominant and non-dominant CSTs. For simplicity, we will refer to the dominant CST, i.e., from the left hemisphere to the right hand, as DxDx and, analogously, SnSn and DxSn for the intra-side non-dominant and inter-side CSTs, respectively. 

In the case of changes between pre- and post-Faremus, we will test whether there is a correlation between the changes induced by Faremus treatment and the amelioration of fatigue symptoms.

## 2. Methods

All methods were carried out in accordance with the Declaration of Helsinki. The Ethics Committee Lazio1—San Camillo Forlanini approved the experimental protocols (023/CE Lazio1, 11 January 2016). All patients signed the informed consent form before their enrollment. 

### 2.1. Study Design

The present proof-of-concept study focuses on the mechanisms of action of Faremus, while its efficacy tested via two independent double-blind, sham-controlled, crossover randomized controlled trials (RCTs) has been already published [26,27]. The RCT outcome was the fatigue level reduction as measured by the modified fatigue impact scale (mFIS), a validated questionnaire with 21 items assessing chronic fatigue along with the physical, cognitive, and psychosocial dimensions [54].

The present study, ancillary to the main RCTs on the efficacy of Faremus in relieving MS fatigue, assesses the effects of Faremus on the CST homology (Figure 1). We estimated the similarity of left and right MEP morphologies before and after Faremus.

The TMS protocol was executed in baseline (T0; pre-Faremus, the day of the first tDCS application and before the stimulation) and post-Faremus (T1; in 8 patients, TMS was executed on the fifth day, waiting 4 h after the last tDCS stimulation, and in 2 patients, it was executed on the following Monday, i.e., 7 days after T0). Each patient underwent the collection of the TMS-induced MEPs from the two body sides (see Figure 1 and Figure 2). 

Patients were recruited if diagnosed with relapsing–remitting MS according to McDonald’s criteria [55] and complied with the eligibility criteria as follows.

Inclusion criteria were as follows:Absence of clinical or radiological evidence of disease activity (NEDA) for at least 3 months preceding the study;Low degree of disability as estimated by Expanded Disability Status Scale (EDSS, Kurtzke 1983) < 2.5;Fatigue as estimated by mFIS > 30.Exclusion Criteria were as follows:Current or prior (within less than 12 weeks before enrolment) exposure to psychotropic drug(s) (antidepressant, anxiolytic, antipsychotics, anticonvulsants, and myorelaxant drugs);Coexistence of other condition(s) potentially associated with fatigue (i.e., anemia and pregnancy);Current or prior (within less than 4 weeks before enrolment) exposure to anti-fatigue products;History of epilepsy.

The neurologist collected the clinical history that included the disease duration and annual relapse rate, EDSS, and Beck Depression Inventory (BDI).

#### Faremus Treatment (5-Day Anodal tDCS with Personalized S1 Electrode)

As detailed in [26,27], an individualized electrode (regional personalized electrode, RePE) was shaped from the brain anatomical MRI of each patient to fit the central sulcus and target the whole-body somatosensory representation areas (Figure 1). For 5 consecutive days, a RePE anode was properly positioned via neuronavigation [SofTaxic Neuronavigation System ver.2.0 (www.softaxic.com, E.M.S., Bologna, Italy)] over the central sulcus (5 mm anteriorly, 15 mm posteriorly), and tDCS was applied for 15 min per day, delivering 1.5 mA current across the RePE anode (35 cm^2^) and a rectangular occipital cathode centered on the Oz of the electroencephalographic international system, with the long side on the longitudinal direction (10 × 7 cm^2^).

### 2.2. MEP Collection and Analysis

#### Stimulation and Recording Setup

The examined subject laid on a comfortable armchair in a quiet room. The muscular signals (electromyogram, EMG) of the opponens pollicis (OP) of the right and left hands were sensed by two surface electrodes (2.5 cm apart) in a belly–tendon montage. We performed single-pulse TMS through a standard focal coil connected with a SuperRapid module (The Magstim Company Ltd., Whitland, UK). For each subject, we searched for the coil position evoking optimal MEP from the contralateral OP, and we assessed the motor resting threshold (RMT)—defined according to international standards as the intensity eliciting MEPs in the 50 microV amplitude scale in about 50% of 16 consecutive trials [56]. TMS was applied at an intensity adjusted to 120% of the RMT. TMS stimuli were elicited with an interpulse interval randomly changing between 4 s and 6 s collecting about 20 repetitions for each side [57,58]. The available MEPs resulted an average of 18 per side.

### 2.3. MEP Morphology Similarity

We used the Fréchet distance estimate implemented in Matlab [59]. The Fréchet distance estimates the minimum cord length sufficient to join two points travelling forward along two distinct curves, considering the rate of travel for either point not necessarily uniform.

To estimate the individual similarity between the two CST hemi-bodies, we evaluated the Fréchet distances between each of the 18 right and 18 left OP MEPs, obtaining 324 DxSn MEP Fréchet distances. Similarly, the intra-sided estimates consisted of $\left(\begin{array}{c} n\\ k \end{array}\right)$ with n = 18 and k = 2, resulting in 153 Fréchet distances (for each DxDx and SnSn).

### 2.4. Statistical Analysis 

Given the large number of values, we did not rely on the Shapiro–Wilk test for normality. By examining the distribution, we found huge numbers of outliers, and, for this reason, we transformed the original Fréchet distance values using the natural logarithmic function. The amelioration of Gaussian fitting was not satisfactory; thus, analyses were carried out by non-parametric Wilcoxon matched paired signed-rank tests.

We reported a result for the significance of the effect at *p* < 0.050.

To be consistent with the current literature, we also evaluated the latency and amplitude of MEPs in the two sides of the body and the inter-lateral difference, focusing particularly on the variance of intra- and inter-lateral amplitude, testing the hypothesis that Faremus will modify the homology between the two CSTs by modulating MEP similarity in a physiological direction.

Statistical analysis was performed using SPSS 27.

### 2.5. Data Availability

MEPs, Fréchet algorithms, personal, and clinical anonymized data will be available upon request.

## 3. Results

Ten subjects with MS suffering from fatigue entered the present study. The fatigue levels reduced after with respect to before Faremus (Table 1, two-tailed *t*-test for paired samples, t(9) = 2.556, *p* = 0.031) with a 27% amelioration on average, as expressed by the mFIS percentage change, i.e., the difference between the mFIS scores before and after Faremus divided by the mFIS score before it (Table 1). 

### 3.1. Faremus Effects on MEP Morphology Similarity as An Index of the Two CSTs’ Homology

The non-parametric Wilcoxon one-sided signed-ranks test (Pre > Post) indicated that the inter-side DxSn Fréchet distance before Faremus (mean rank = 5.414) was higher than after Faremus (mean rank = 5.061, Z = 1.886, *p* = 0.032; Figure 3A). 

Intra-side comparisons did not show that the Fréchet distance before Faremus was higher than after Faremus, either in DxDx (Z= 1.478, *p* = 0.080) or SnSn comparison (Z= 0.255, *p* = 0.423; Figure 3B,C).

### 3.2. Relationships between MEP Shape Similarities Changes and Fatigue Amelioration

The changes in inter-body-side homology similarity did not show a significant correlation with the changes in mFIS (*p* >0.05, consistently). 

### 3.3. MEP Amplitude and Latency

No effect was observed in terms of MEP amplitudes and latencies either as differences between the two body sides or as an effect of Faremus treatment (Table 2). 

## 4. Discussion

As a key result, Faremus induced a change in the physiological direction of the homology between the two corticospinal tracts, as estimated by the similarity of the MEP morphology via the Fréchet distance. Interestingly, the modifications induced by Faremus were more evident in the inter-side relationship than dominant and non-dominant intra-sides.

### 4.1. CST Asymmetries and Fatigue

Very recently, the anatomical CST correlation with fatigue in MS patients has been assessed via anatomical connectivity mapping (ACM) based on diffusion MRI tractography [45]. Unexpectedly, with respect to healthy volunteers, MS patients bilaterally displayed altered higher ACM. In terms of the relationship with fatigue, while inter-hemispheric asymmetries outside the CST did not scale with individual symptoms, the higher the ACM values in the left relative to the right CST, the more severe the symptoms were. In agreement with higher CST morphological asymmetries in MS fatigue with respect to healthy controls, our functional assessment evidenced much higher inter-body-side CST asymmetries with Fréchet distances ranging around double values than the distribution in healthy volunteers [53]. 

Similar behavior has also been observed in other neuropathological conditions. Three months of paretic arm robotic rehabilitation changed the inter-hemispheric coupling of somatosensory homologs [60], which distributed across a wide range before robotic rehabilitation but in a narrow range after treatment. The authors speculated that the inter-hemispheric connectivity found at the end of the rehabilitation program corresponded to a more functionally efficacious and "physiological" condition since patients reached a clear motor improvement by robotic rehabilitation. Similarly, in the present case, we can speculate that a better physiological condition consequent to an amelioration of fatigue induced by Faremus can reflect into a more physiological state of the CST homology. 

The asymmetries of the CST excitability in MS patients were recently evaluated in terms of TMS resting and active motor thresholds and intracortical inhibitory efficacy via the cortical silent period. Intracortical inhibition alterations and increased asymmetry of excitability were significantly correlated with more severe fatigue symptoms. Interestingly, altered inter-lateral asymmetries evidenced lost dominance with respect to the non-dominant side [37]. Notably, in the present investigation, the smaller intra-side distance with respect to the inter-side one previously observed in a healthy sample was lost pre-Faremus for the dominant CST (one-tailed *t* test, t(9) = 1.201, *p* = 0.0130), Figure 4A) and regained this property post-Faremus (t(9) = 2.736, *p* = 0.011), Figure 4D). In accordance with the alterations with MS fatigue prominent in the dominant side, the dynamics of neuronal electrical activity, neurodynamics, lost the physiological greater complexity in M1 compared with S1 in the dominant hemisphere [11].

### 4.2. CST Homology Modified by Faremus S1 Neuromodulation

Faremus personalized S1 neuromodulation, giving rise to a fatigue amelioration, changed the homology of the two CSTs. 

The efficacy of Faremus is supported by two previous sham-controlled randomized trials [26,27]. Even in the present subgroup, the improvement in fatigue levels at 27% on average can be considered clinically significant, as typically 20% is indicated as the threshold to identify responders to a given intervention. 

Concerning the speculation on the possible effects of Faremus intervention on CST homology, we integrate the present with previous observations that Faremus treatment considerably affected the neurodynamics that changed more strongly in S1, the target of the neuromodulation, than in M1. Nevertheless, in terms of functional connectivity, the changes were mostly evident in the connection between homolog left and right primary motor areas [11]. Here, we observed that Faremus changed in a physiological direction the similarity between the two homolog CSTs. These effects on the CSTs induced by Faremus S1 neuromodulation highlight how sensory and motor counterparts are the two sides of the same coin. A suggestive example of such an inextricable relationship can be found in a TMS study, where MEPs were simultaneously recorded from two hand muscles innervated by the same nerve, before and after the anesthetic block of the nerve supplying the somatosensory inflow from only one of the two, demonstrating that the muscle with sensory deprivation selectively lost part of its motor representation [61]. Extending the observation to behavioral effects, a paradigmatic example of the critical somatosensory role in motor execution comes from the selective perturbation of goal-directed reaching movements as a consequence of selectively blocking by genetic manipulation the sensory inflow from the forelimb [62]. 

### 4.3. Central More Than Peripheral Origin of MS Fatigue

In a group of MS patients suffering from a wide range of fatigue, we found that the alterations of the cortico-muscular coherence correlated with fatigue severity, while this was not the case for the behavioral alterations detectable in motor execution [63]. It is interesting to complement the Faremus-induced improvement of fatigue symptoms associated with changes in cortical neurodynamics and intracortical functional connectivities [11] with the lack of association with effects on the CST found here. These pieces of evidence strengthen the role of central mechanisms more than peripheral–behavioral ones at the origin of MS fatigue and lead us to speculate that the Faremus-induced changes observed in MEPs’ shape similarity are a consequence of the changes of central mechanisms at the origin of fatigue. 

The authors who studied the corticospinal excitability by TMS and peripheral electric stimulation before and after a fatiguing task arrived at similar conclusions in people with MS, reporting a more central than peripheral origin of behavioral fatigue mechanisms [64].

A possible reason for the lack of relationships with clinical changes can derive from the non-ecologic TMS. The TMS implies supra-physiological synchronizations, which are excellent in assessing the circuit propagation velocities and fiber impairments—relevant for clinical evaluation in many pathological conditions including MS, carpal tunnel syndrome, etc.—but less sensitive to neuronal networking alterations subtending symptoms not secondary to major damages and tissue degeneration. 

### 4.4. MEP Shape vs. Amplitude

We found that the assessment—via Fréchet distance– of the MEP morphology was sensitive to Faremus-related changes. Instead, the MEP amplitudes did not change by the treatment. We previously observed that in MS patients, the evoked activity morphology is sensitive to subtle changes induced in the central circuitry more than the response amplitude [10]. More recently, we found that the somatosensory evoked potential morphology can be exploited to enhance the monitoring sensitivity to cerebral blood flow reduction than the SEP amplitude [47]. With remarkable relevance, the morphology of MEPs sensitive to homology features that do not emerge from MEP amplitudes appeared in addition to pathological conditions even in healthy states [53]. 

### 4.5. Limitations of the Present Work 

The present exploratory work presents limitations that can be addressed through future research.

The CST homology was here observed after the treatment, and a longitudinal assessment of its behavior along with the re-emergence of fatigue would be interesting.

Data in healthy individuals were provided in a previous work [53] in a consistent group of people with similar age but collected in a different laboratory in the frame of an independent experiment that collected data of the first dorsal interosseous muscle. 

The CST homology investigation, ancillary to Faremus clinical trials, was executed in a subgroup of patients undergoing only the real treatment. It would be advisable to carry out the same investigation in a larger group of MS subjects with a sample size comparable with that of the two Faremus RCTs and perform a direct comparison with a double-blind, sham-controlled, crossover design.

We analyzed the CST comparing the dominant vs. non-dominant hemi bodies without correlating with the handedness level, as assessed, for example, by the scoring of the Edinburgh Inventory. Our analysis strategy is based on the concept that left-handers, in addition to being about 10% of the population, do not invert brain organization with respect to right-handers, and handedness-dependent dominance brain asymmetries are more pronounced in right-handed than in left-handed people. Nevertheless, enrolling larger samples of people would be interesting to take this variability into account. Notably, the commutative property holds, as for every distance measure, for the Fréchet distance, so that evaluations do not depend on which side is the dominant one.

## 5. Conclusions

The present data strengthen the knowledge of the relevance of sensorimotor feedback for brain physiology since S1 neuromodulation, which ameliorated fatigue symptoms, changed the homology between the dominant and non-dominant CSTs. The present results support also a central more than peripheral–behavioral mechanism as the origin of MS fatigue. All in all, the results discussed here are in line with the emergence of new measures that, sensitive to network characteristics, provide new insights into the mechanisms of neuronal organization underlying relevant symptoms.

## Figures and Tables

**Figure 1 brainsci-13-00574-f001:**
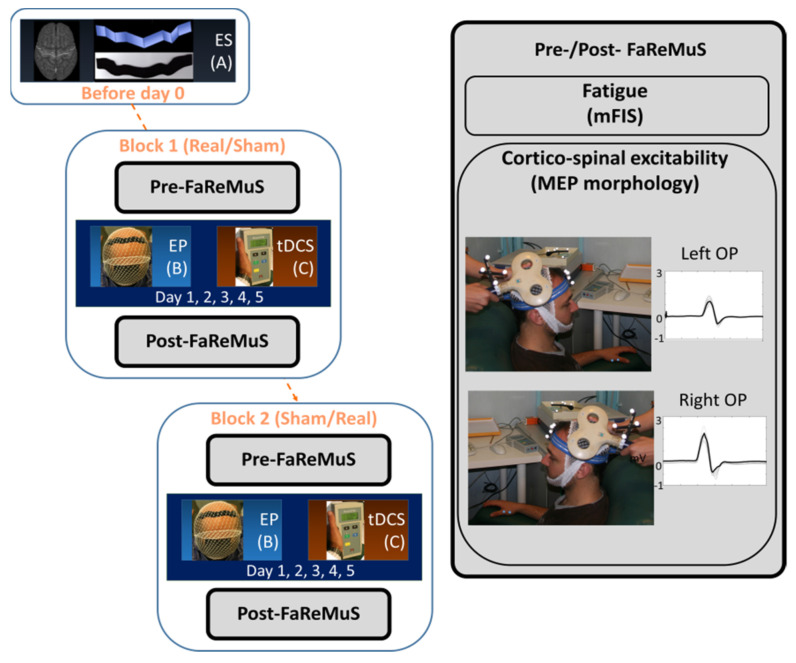
Experimental procedure in each fatigued person with MS undergoing Faremus treatment. **Left** from **top**: Individual brain MRI-based personalized electrode shaping [ES (A)] performed once before tDCS sessions. In blocks 1 and 2 (equal, but the 5 −day stimulation is either real or sham based on random assignment), after the data collection (pre-Faremus), the electrode positioning [EP, (B)] is executed before each 15 min stimulation [tDCS, (C)] of each of the 5 days of treatment. On the day of the last tDCS, we performed the data collection (post-Faremus). **Right**: Data collection (pre − and post −Faremus) includes the fatigue level assessment (modified fatigue impact scale, mFIS) and the collection of left/right opponens pollicis (OP) motor-evoked potential (MEP) by TMS, used for corticospinal tract excitability and inter-side homology assessment.

**Figure 2 brainsci-13-00574-f002:**
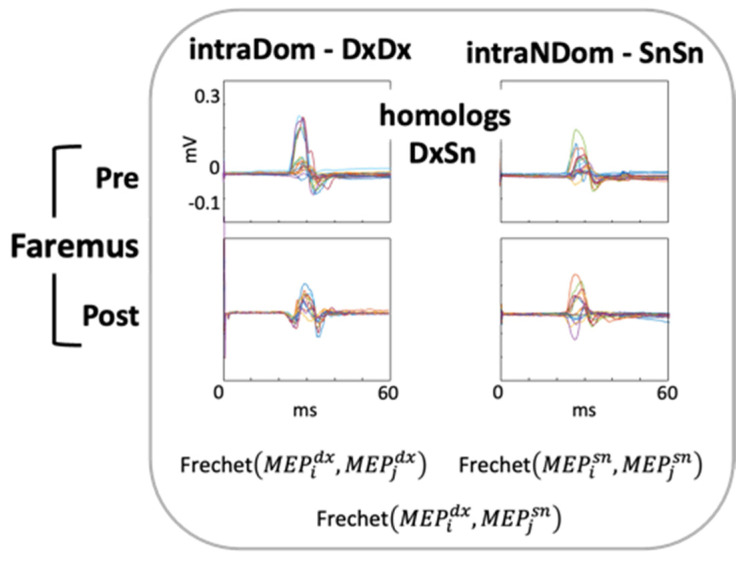
Fréchet distance to assess MEP morphology similarity. Before (Pre) and after (Post) Faremus personalized neuromodulation treatment against MS fatigue; superimposition of about 20 MEPs (colored lines, amplitude scale equal for all conditions) collected in one exemplificative patient. Fréchet distances between left and right MEP morphologies measured the homology of dominant and non-dominant CSTs (homologs, DxSn), and those between the MEP of the same side measured the hemi-lateral pattern recruitment variability (intraDom, DxDx and IntraNDom, SnSn).

**Figure 3 brainsci-13-00574-f003:**
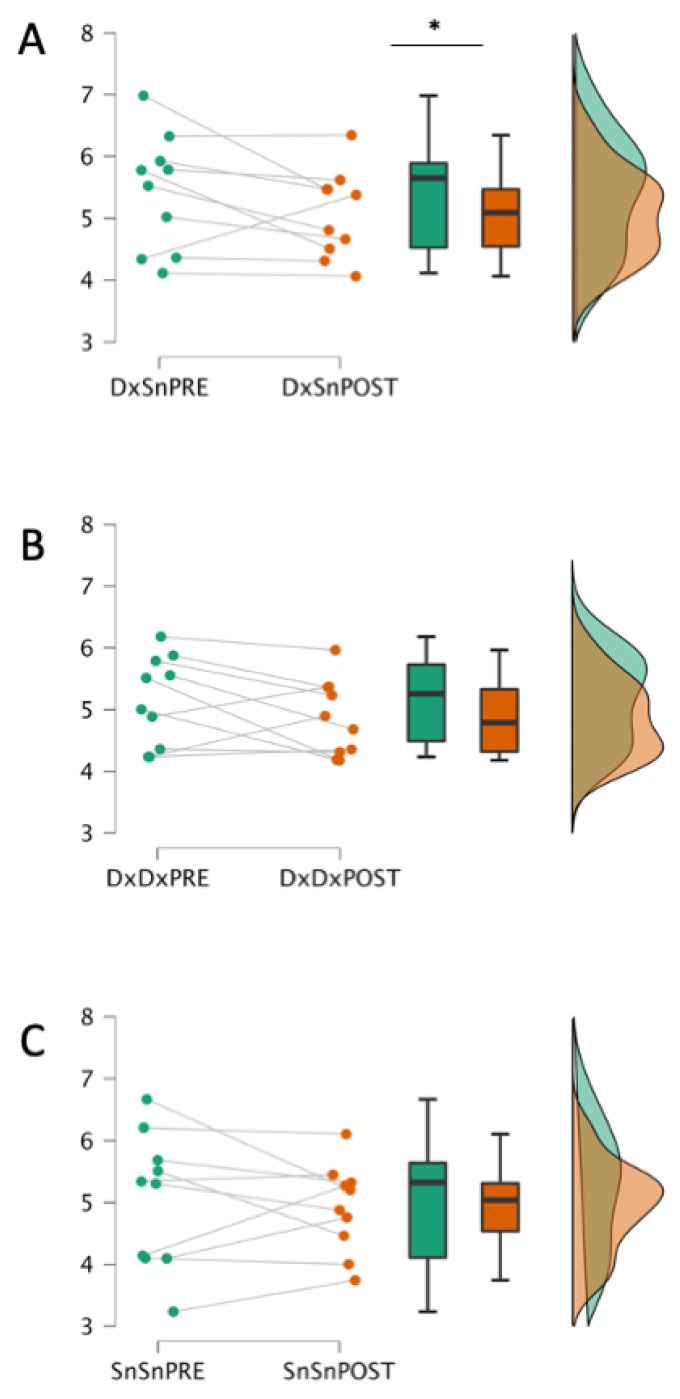
Faremus effects on CST inter-side homology and intra-side recruitment variability. One-sided signed-ranks test on inter-side DxSn (**A**) and intra-side DxDx (**B**); SnSn (**C**) MEP Fréchet distances before (PRE) and after (POST) Faremus treatment. Scale values in logarithmically transformed mV. Only the inter-side homology (**A**) displayed a significant reduction after with respect to before Faremus (* *p* < 0.05).

**Figure 4 brainsci-13-00574-f004:**
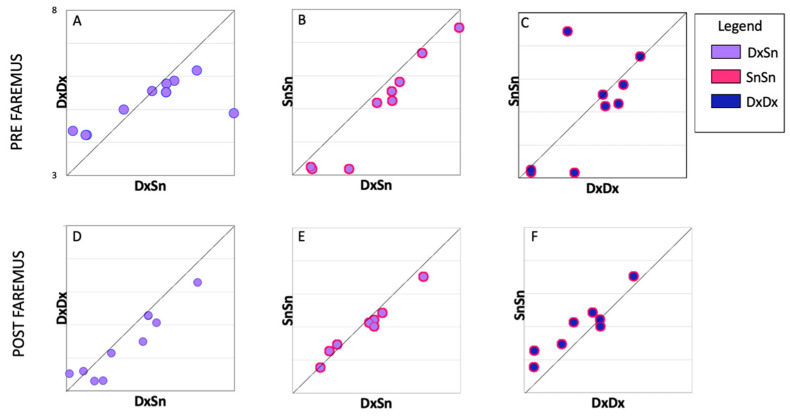
Individual comparison of inter-side homology and intra-side recruitment variability. Boxplots of individual Fréchet distance between inter-lateral MEP morphologies (DxSn) (mean of around 324 pairs for each person, x-axis) vs. intra-lateral distances (mean of about 153 Fréchet values for each person, y-axis) before (PRE, top) and after (POST, bottom) Faremus treatment. All boxes have the same scale 3–8 (ln mV) in x- and y-axes. Inter-lateral distance is higher than dominant right intra-lateral distance after Faremus (**D**) while not before (**A**) and higher than non-dominant left intra-lateral distance both before (**B**) and after (**E**) Faremus. The dominant intra-lateral recruitment variability did not differ from the non-dominant either before (**C**) or after (**F**) Faremus treatment.

**Table 1 brainsci-13-00574-t001:** MS patient clinical profile and fatigue with Faremus.

		Mean *Median*	SD [Min-Max]
Patients’ Personal and Clinical Profile	Sex 9 Women, 1 Man		
	Age (years)	35.3	9.3
	Disease duration (years)	3.2	2.0
	Annual relapse rate	*0*	[0,1]
	Expanded Disability Status Scale	*0*	[0–2.5]
	Beck Depression Inventory	8.5	0.7
Modified Fatigue Impact Scale	Pre Faremus	47 *	15
	Post Faremus	38 *	22
	%	27	33

Mean and standard deviation (SD) of personal and clinical variables. When the value distribution was not Gaussian, instead of the mean and SD, we reported the median and variability range. %: Percentage change in each patient is the (pre-post)/pre mFIS value. Asterisks refer to significant change after than before Faremus.

**Table 2 brainsci-13-00574-t002:** MEP features.

	MEP Latency		MEP Amplitude	
	MEP dx	MEP sn	MEP dx	MEP sn
Pre Faremus	27.1	27.3	324	445
	(7.2)	(3.2)	(793)	(651)
Post Faremus	27.6	27.5	291	329
	(6.9)	(4.2)	(925)	(378)

Mean and standard deviation of inter-side CSTs’ homology estimated as Fréchet distances between left and right MEP latency (ms) and amplitude (µV). Data are presented before (Pre) and after (Post) the personalized neuromodulation Faremus against MS fatigue.

## Data Availability

The data presented in this study are available on request from the corresponding author.

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
