# Peer review of "Effects on Corticospinal Tract Homology of Faremus Personalized Neuromodulation Relieving Fatigue in Multiple Sclerosis: A Proof-of-Concept Study"

_brainsci, 2023, doi:10.3390/brainsci13040574_

Round 1
Reviewer 1 Report
This is a very interesting and well written paper. However, I have some comments and suggestions which may improve the quality of this manuscript.
Please provide a more recent definition of fatigue.
The introduction should be more critically. is there sufficient evidence that tDCS can increase cortical excitability?
Are MEPs correlated with fatigue measures?
Justify the absence of a control group.
Include a paragraph about limitations and future studies.
Author Response
This is a very interesting and well written paper. However, I have some comments and suggestions which may improve the quality of this manuscript.
Please provide a more recent definition of fatigue.
- We thank the Reviewer for this observation. In a recent review ( doi: 10.3389/fneur.2022.813965. eCollection 2022.), Ayache et al. advocate for the use of the definition provided by the MS Council for Clinical Practice Guidelines defining MS fatigue as “a subjective lack of physical and/or mental energy that is perceived by the individual or caregiver to interfere with usual and desired activities”. We updated the reference accordingly.
International Classification of Functioning, Disability and Health (ICF) developed a Core Set specifically to assess functioning, activity, and disability in MS. ICF core set is intended as an international standard ofwhat to measure and report but not how to measure it. Among the category “Body functions”, fatigue stands out. Despite the lack of agreement on how to assess fatigue, in our experience we would integrate the reported experience of this symptom with a quantification of fatigue levels as above a ‘clinical relevant threshold’ (with some oscillation internationally) by means of well-known and widely used standardized tests (e.g Modified Fatigue Impact Scale).
The introduction should be more critically. is there sufficient evidence that tDCS can increase cortical excitability?
- We thank the Reviewer for focusing on this important aspect.
While wide work is ongoing to hone precise effects of tDCS, the about 5000 citations of the following two works (Nitsche et al. 2008 and Brunoni et al. 2012) support the inconvertible evidence of tDCS ability to modify temporally (from minutes to hours) or more durably (up to months) the target excitability. We updated the references including these literature milestones about tDCS effects.
Nitsche MA, Cohen LG, Wassermann EM, Priori A, Lang N, Antal A, Paulus W, Hummel F, Boggio PS, Fregni F, Pascual-Leone A. Transcranial direct current stimulation: State of the art 2008. Brain Stimul. 2008 Jul;1(3):206-23. doi: 10.1016/j.brs.2008.06.004. Epub 2008 Jul 1. PMID: 20633386.
Brunoni AR, Nitsche MA, Bolognini N, Bikson M, Wagner T, Merabet L, Edwards DJ, Valero-Cabre A, Rotenberg A, Pascual-Leone A, Ferrucci R, Priori A, Boggio PS, Fregni F. Clinical research with transcranial direct current stimulation (tDCS): challenges and future directions. Brain Stimul. 2012 Jul;5(3):175-195. doi: 10.1016/j.brs.2011.03.002. Epub 2011 Apr 1. PMID: 22037126; PMCID: PMC3270156.
Romero Lauro LJ, Rosanova M, Mattavelli G, et al. TDCS increases cortical excitability: direct evidence from TMS-EEG. Cortex. 2014;58:99-111. doi:10.1016/j.cortex.2014.05.003
Are MEPs correlated with fatigue measures?
- Again, we thank the Reviewer for this consideration.
MEP investigations, typically parametrized by their amplitude, were extendedly used to assess the pathophysiology of MS fatigue, although their amplitude has never been associated to fatigue severity scales’ scores. In agreement, also in the present case, we exploited MEP not to assess fatigue severity but instead to assess the central-peripheral recruitment features in MS fatigued patients.
Justify the absence of a control group.
- The present is a Proof of Concept study not primarily intended to validate the efficacy of Faremus treatment but to explore the effects of the treatment on MEP similarity grounding on our working hypothesis that MS fatigue patients display an electrophysiological profile compatible with local alterations of physiological excitatory-inhibitory mechanisms, secondary or giving rise to fatigue itself, that also reflect into imbalances of the inter-hemispheric functional relationship between sensorimotor regions both at rest and during movement execution.
Nevertheless, we previously collected and described the measure used here for assessing the CSTs’ homology in a group of healthy volunteers, comparable in age to the present group of MS patients. We used the knowledge derived from that work, to discuss the present results:
‘Pagliara MR, Cecconi F, Pasqualetti P, Bertoli M, Armonaite K, Gianni E, Grifoni J, L'Abbate T, Marinozzi F, Conti L, Paulon L, Uncini A, Zappasodi F, Tecchio F. On the Homology of the Dominant and Non-Dominant Corticospinal Tracts: A Novel Neurophysiological Assessment. Brain Sci. 2023 Feb 7;13(2):278. doi: 10.3390/brainsci13020278. PMID: 36831821; PMCID: PMC9954672.’
Include a paragraph about limitations and future studies.
- We thank the Reviewer for this suggestion that we followed including a new devoted paragraph in the revised version.
Reviewer 2 Report
Bertoli et al. presented an interventional study using 5-day tDCS on improving fatigue in patients with multiple sclerosis. The results showed the revision of interhemispheric imbalance (MEP) after the intervention.
Abstract:
The study design should be clearly mentioned in the abstract. In addition, any outcome related to fatigue that was used in the study should be mentioned.
Introduction:
NIBS for multiple sclerosis-related fatigues usually targets M1 or DLPFC. The rationale for using S1 as the target needs to be clearly mentioned in the introduction.
In addition, how the balance between two hemispheres is correlated with fatigue is largely not explained.
I do not understand why the authors select M1-MEP to evaluate the effect of S1 tDCS.
Methods:
The study design was not clear.
If an individual MRI was used, please report the scanning protocol and parameters.
Why the total trials were 16 when determining RMT?
Results:
Table 1: Heading – mean Median/SD range. It is not clear which value the authors intend to report in the table.
Due to the variability of MEP, usually, multiple trials of MEP are averaged and then used as endpoint. I do not very understand the reason why the authors decided to use the Frechet distance for intra-MEP evaluation.
What were the units for the Y axis in Figure 3?
Please label significant pairs in Tables 1 and 2.
To sum up, I can’t understand the rationale of the current study – using S1 tDCS for modulating MEP over the bilateral M1, and how the neurophysiological findings can be connected to the improvement of fatigue. Due to the limitation of the lack of a sham-controlled group, it is very hard to elaborate on the treatment mechanism.
Author Response
Bertoli et al. presented an interventional study using 5-day tDCS on improving fatigue in patients with multiple sclerosis. The results showed the revision of interhemispheric imbalance (MEP) after the intervention.
Abstract:
The study design should be clearly mentioned in the abstract. In addition, any outcome related to fatigue that was used in the study should be mentioned.
- We thank the Reviewer for this criticism, allowing us to better clarify the present study design in the revised version. In fact, we tried to better clarify that the present is an ancillary study to deepen understanding of the mechanisms subtending MS fatigue and its relief via Faremus treatment. In this respect, the clinical outcome of previously published randomized clinical trials (RCT, fatigue levels reduction as assessed by mFIS) is not the focus of the present study. Nevertheless, we reported the mFIS values pre- and post-Faremus related to the present subgroup to assess the relationship with the central-peripheral recruitment properties changes after Faremus.
Introduction:
NIBS for multiple sclerosis-related fatigues usually targets M1 or DLPFC. The rationale for using S1 as the target needs to be clearly mentioned in the introduction.
- We thank the Reviewer for this consideration. We extended in the revised introduction the concepts that led to the implementation of the Faremus intervention targeting bilateral whole body S1 among other available neuromodulation treatments.
In addition, how the balance between two hemispheres is correlated with fatigue is largely not explained.
- We thank the Reviewer for drawing attention to this relevant aspect. We expanded in the revised text the notion that the dynamic interplay between homologous cortical areas is a critical element for a proper brain functioning either during task execution and at rest and we enhanced the description of correlates with fatigue levels.
I do not understand why the authors select M1-MEP to evaluate the effect of S1 tDCS.
- We extended in the revised introduction that CSTs are not selectively originating from M1 but includes also about 30% of fibers coming from S1 cortical areas. Moreover, the work by Porcaro et al. (2019) tested the working hypothesis that Faremus modified the functional organization of the brain networks involved in MS-related fatigue contributing to its efficacy, also deepening the understanding of the phenomena subtending fatigue and its relief. The investigation showed that before the treatment, the neurodynamic was more distorted in S1 than M1. Nevertheless, in terms of functional connectivity the alteration was more evident between the homologs M1s than S1s. These alterations reverted to normal after Faremus, with a positive correlation between the amelioration of fatigue levels.
We believe that these pieces of evidence support the hypothesis of the present investigation that Faremus S1 neuromodulation treatment can change the central-peripheral recruitment properties, in particular for homology relatioship.
Methods:
The study design was not clear.
- We extended in the revised methods section that the present investigation represents an ancillary study of the main RCTs on the efficacy of Faremus in relieving MS fatigue. Here we exploit the original design which included collecting MEP before and after the 5-days Faremus intervention in a subgroup of patients to test the hypothesis that it modifies, in physiological direction, the homology between the two CSTs.
If an individual MRI was used, please report the scanning protocol and parameters.
- We introduced in the revised methods the information that a single anatomical MRI was enough to derive the individual electrode shape. The detailed procedure to obtain the electrode fitting the individual central sulcus cortical folding is well described in the clinical trials (Tecchio 2014, Cancelli 2018) and other devoted manuscript (Cancelli et al Front Neurosci 2018). Since any anatomical MRI is enough to derive the electrode shape, we provide here for your reference the information of the MRI protocol which has been used (MPRAGE). It is part of a devoted protocol for MS monitoring and volumetric assessments, providing much more information (Tomasevic at al 2013, Zito et al 2014) than the simple one of the central sulcus shape required to realize the electrode.
Brain MRI was performed with a 1.5 T scanner (Achieva, Philips Medical Systems, Best,The Netherlands), provided with 33 mT/m gradient amplitude, online 2D/3D geometric distortion correction and an eight-channel SENSE coil. T1 3D Fast field-echo (MPRAGE) were acquired by the following protocol:
Field of view (FOV), mm |
240 |
Repetition time (TR), ms |
8,6 |
Echo time (TE), ms |
4 |
Inversion time (TI), ms |
— |
Flip angle (FA) |
8 |
Slice thickness, mm |
1,25 |
Slice gap, mm |
0 |
Nr of slices |
170 |
Acquisition time, min |
6,23 |
Plane |
sagittal |
Matrix |
192x192 |
Number of acquisitions |
2 |
- Tomasevic L, Zito G, Pasqualetti P, Filippi M, Landi D, Ghazaryan A, Lupoi D, Porcaro C, Bagnato F, Rossini P, Tecchio F. Cortico-muscular coherence as an index of fatigue in multiple sclerosis. Mult Scler. 2013 Mar;19(3):334-43. doi: 10.1177/1352458512452921. Epub 2012 Jul 3. PMID: 22760098.
- Zito G, Luders E, Tomasevic L, Lupoi D, Toga AW, Thompson PM, Rossini PM, Filippi MM, Tecchio F. Inter-hemispheric functional connectivity changes with corpus callosum morphology in multiple sclerosis. Neuroscience. 2014 Apr 25;266:47-55. doi: 10.1016/j.neuroscience.2014.01.039. Epub 2014 Jan 29. PMID: 24486438; PMCID: PMC4143150.:
- Cancelli A, Cottone C, Giordani A, Asta G, Lupoi D, Pizzella V and Tecchio F (2018) MRI-Guided Regional Personalized Electrical Stimulation in Multisession and Home Treatments. Front. Neurosci. 12:284. doi: 10.3389/fnins.2018.00284
Why the total trials were 16 when determining RMT?
- The procedures for resting motor threshold assessment are worldwide consolidated (Rossi et al 2009 with 3600 citations, Rossini et al. 1994). They indicate to increase the stimulus intensity progressively at steps of 5% of the stimulator output until reaching a level which induces reliable (usually around 100 ~micronV) MEPs in about 50% of 10-20 consecutive stimuli. We updated the revised text accordingly.
Results:
Table 1: Heading – mean Median/SD range. It is not clear which value the authors intend to report in the table.
- We updated the format of the table. When the values distribution was not fitting a gaussian, instead of the Mean and Standard deviation we reported the Median and variability range.
Due to the variability of MEP, usually, multiple trials of MEP are averaged and then used as endpoint. I do not very understand the reason why the authors decided to use the Frechet distance for intra-MEP evaluation.
- In the present study, we introduce also for MEP, a concept previously introduced for somatosensory evoke fields (SEF) and potentials (SEP), i.e that the morphology of the response is able to carry information about the recruitment pattern of the involved circuit. We chose Fréchet distance as a more sensitive measure for comparing pattern morphologies as we have done already in a similar investigation in a sample of healthy individuals comparable in age to the present MS group (Pagliara et al 2023). Notably, we also evaluated the standard endpoint of the multiple MEP trials averaged and their amplitude quantified.
What were the units for the Y axis in Figure 3?
- We thank the reviewer for this note. The Fréchet distance is in the same MEP unites (mV). We included this information in the caption of Figure 3.
Please label significant pairs in Tables 1 and 2.
- In Table 1, we integrated the significant change in mFIS. The mean improvement in fatigue levels after Faremus (27% on average) is considered clinically significant (>20%).
With regards to Table 2 reporting MEP features (latency and amplitude), no effect was observed either as differences between the two body sides or as an effect of Faremus treatment.
To sum up, I can’t understand the rationale of the current study – using S1 tDCS for modulating MEP over the bilateral M1, and how the neurophysiological findings can be connected to the improvement of fatigue. Due to the limitation of the lack of a sham-controlled group, it is very hard to elaborate on the treatment mechanism.
- Following the indications provided by the Reviewer, we believe we have improved the manuscript by clarifying that this is an ancillary investigation focusing on changes in central-peripheral functional indices subsequent to a personalised neuromodulation intervention whose efficacy in relieving MS fatigue has been consolidated by 2 randomised controlled trials and meta-analysis.
Round 2
Reviewer 1 Report
I disagree with your definitions of fatigue [3,4].
I suggest to use definitions provided by Enoka et al. 2021 or "The decrease in physical and/or mental performance that results from changes in central, psychological, and/or peripheral factors" (Rudroff et al. 2016).
Furthermore, I suggest to add -A proof of concept study-to the title.
Author Response
Aware of the difficulty of finding a widely accepted definition of fatigue, we have willingly adopted the one proposed by the Reviewer.
We also gladly introduced the specification on the nature of the study in the title.
Reviewer 2 Report
The authors have addressed my previous concerns.
Author Response
Very good.